# Fieldwork and Field Trials in Hospitals: Co-Designing A Robotic Solution to Support Data Collection in Geriatric Assessment

**Karine Lan Hing Ting [1,\*], Dimitri Voilmy [1], Quitterie De Roll [2], Ana Iglesias [3] and Rebeca Marfil [4]**

1. Living Lab ActivAgeing, UR LIST3N, Université de Technologie de Troyes, 10004 Troyes, France; dimitri.voilmy@utt.fr
2. CRRF COS Pasteur, 10000 Troyes, France; qderoll@fondationcos.org
3. Departamento Informática, Universidad Carlos III de Madrid, 28911 Madrid, Spain; aiglesia@inf.uc3m.es
4. Departamento Tecnologia Electronica, Universidad de Málaga, 29071 Málaga, Spain; rebeca@uma.es
* Correspondence: karine.lan@utt.fr

**Featured Application: The paper presents a use case of a robotic assistant for geriatric evaluation procedure at the hospital. Design and evaluation of assistive social robots for clinical routine is extensively discussed. The design and methodological questions raised include: human-robot interaction, technology and older adults, accessibility and human-centered design, robot navigation, pilot experimentation, healthcare technology.**

**Abstract:** *Comprehensive geriatric assessment* (CGA) is a multidimensional and multidisciplinary diagnostic instrument that helps provide personalized care to older adults by evaluating their state of health. This evaluation is based on extensive data collection in order to develop a coordinated plan to maximize overall health with aging. In the social and economic context of growing ageing populations, medical experts can save time and effort if provided with interactive tools to efficiently assist them in doing CGAs, managing either standardized tests or data collection. Recent research proposes the use of social robots as the central part of this optimization of clinicians' time and effort. This paper presents the first and last steps of the research made around the design and evaluation of the CLARC robot: fieldwork (analysis of needs and practices concerning clinical data management) and field trials (pilot experiment in real-life conditions in a rehab hospital). Based on an extensive literature review of social robotics applications for health and ageing, it discusses the practical and methodological questions raised around how to design and test assistive social robots for clinical routine, and questions the feasibility of an automated CGA procedure.

**Keywords:** *comprehensive geriatric assessment*; social robotics; health data collection; clinical data management; analysis of practice; real-life experimentation

## 1. Introduction

*Comprehensive geriatric assessment* (CGA) is a multidimensional and multidisciplinary diagnostic instrument that helps provide personalized care to older adults, by evaluating their state of health [1]. This evaluation is based on extensive data collection about the frail older person's medical, psychosocial, and functional limitations, in order to develop a coordinated plan to maximize overall health with aging [2]. Being an interdisciplinary effort, it requires the coordination of several clinical professionals [3]. This coordination rests, for a large part, on patient data sharing. Improving the diagnosis, creating correct, customized and proportional therapeutic plans, increasing functional autonomy, and also reducing complications during hospitalizations and mortality, are some of CGA's benefits.

In the social and economic context of growing ageing populations, CGA has the potential to contribute efficiently to frailty prevention. However, the associated drawbacks

are increased costs, resources and data management. The geriatrician from the University hospital collaborating on the project explains: "*The problem today, regardless of the country, is that there is a lack of geriatricians in relation to the needs of the population, and that is a constant. Geriatrics, or geriatricians, can absorb at best 10% of the entire body of older people. We should be able to absorb a greater number of frail patients in order to try to render a service to a greater number of people*".

CGA is usually carried out every 6 months and involves both patients and their relatives. It comprises three different types of activities: clinical interview, multidimensional assessment and customized care plan. First, a clinical interview allows the patient and relatives to discuss the older adults' health problems with the physician. Then, multidimensional clinical tests are performed to evaluate the overall patient status. Finally, taking into account the evidence gathered during the two previous phases and the patient's evolution since the last CGA session, physicians create a personalized care plan. A typical CGA session takes about 3 h of a clinician's time. Some of the activities require the clinical staff to be present, but other ones, particularly the multidimensional assessment, are standard tasks suitable for automation and/or parallelization (see [4,5] for more detail).

Indeed, medical experts can save time and effort if provided with interactive tools to efficiently assist them in doing CGAs, managing either standardized tests or data collection. On the basis of knowledge capitalized since the beginning of the project, the research hypothesis was that a robotic solution would allow a gain in the efficiency of geriatric follow-up: automated data collection, better backup and the sharing of secure data, better management of caregivers' time on high value-added activities, such as the development of the personalized care plan. Discharging part of the CGA on a robot would allow clinicians to focus on activities with more added value, such as deciding, together with the patient and relatives, the appropriate care plan. This can be summarized as follows: clinicians' activities are maximized on essential activities, while repetitive or standardized tasks can be delegated to the robot. This division of tasks is in line with the ethical principles, which have guided this research all the way through, and with the robotic principle [6], which have been applied from the beginning of this research. The main principle is that replacing clinicians and caretakers by robots is not an issue, because precisely, being given the high added value of the activities that they keep, clinicians cannot be replaced by the CLARC(CLARC is the name retained for the robot in line with the project's name: ECHORD++ CLARC (the European Clearing House for Open Robotics Development) project (No. FP7-ICT-601116)") robot. Instead, the activities—that are repetitive, time-consuming and of low added value—are delegated to the robot. The issue is not about "replacing humans" but simply about "delegating to the robot" (for the best). This gain in efficiency would allow better care—which is currently insufficient—for the growing elderly population.

Through the presentation of the first and last steps of the research made around the design and evaluation of the CLARC robot, this paper looks into the feasibility of an automated CGA procedure, and reflexively discusses the insights. It discusses the practical and methodological questions raised around how to design and test assistive social robots for clinical routine. The main insights are both (i) positive user feedback about the gain in efficiency hypothesis or promising technical performance and (ii) a real complexity of human–robot interaction, in terms of human interface experience.

This paper is organized as follows, structuring the discussion around the first and last steps of this research: fieldwork and field trials. The extensive literature review of previous work about social assistive robots is put in parallel with the research hypothesis concerning the added value of such a robot for CGA in terms of efficiency. After the methodological approach (Section 3) the fieldwork is presented (Section 4): the needs and practices concerning clinical data management we analyzed to inform the design of such a robot, that is able to interact efficiently with the patient to gather data and by the clinicians' application, CGAMed, which allows clinical data management. The second main part of

the paper presents the field trials (Section 5): the pilot experiment in real-life conditions in a rehab hospital, which shows good acceptance of the robot and satisfaction from patients.

## 2. Literature Review

Recent research proposes the use of social robots as potential useful tools for CGAs. Social robots are characterized as being able to understand and communicate in a human-like way, allowing them to behave as social actors and be understood as such by their users [7]. Knowledge capitalized since the beginning of the project, including the field work observations presented in this paper about data work, allowed the formulation of this research hypothesis: a robotic solution would allow a gain in efficiency in geriatric follow-up. It would allow automated data collection, the better backup and sharing of secure data, and better management of the caregivers' time, allowing the optimization of clinicians' time and effort on high value-added activities, such as the development of the personalized care plan. This hypothesis is in line with the literature review of social robotics applications for health and ageing, and the evaluation factors that ensure efficient human–robot interaction.

### 2.1. Socially Assistive Robots in Health System

The World Health Organization meeting in 2005 [8] adopted a resolution on e-Health, recognizing the need for information and communication technologies (ICTs) in order to improve the health monitoring and management of the health systems. Nowadays, the use of ICTs and complex systems as robots are used in health centers and hospitals. Assistive robotics (ARs) provide assistance to the patients. For instance, wearable robots or exoskeletons are useful for patients with motor impairments [9]. Rehabilitation robots are useful to patients with visual or motor disabilities, older adults, etc. [10]. Pulmonar lesion diagnosis can now be performed by AR [11].

This paper presents a robotic solution integrated in the health systems which provides assistance to the patients through social interaction. These kinds of robotic solutions are called social assistive robots (SARs). Two categories of SARs can be found for health purposes, providing different services to the users through social interaction: services robots and companion robots [12]. For instance, in mental healthcare, SARs usually provide the services of being a therapeutic play partner or being a coach or instructor, or simply provide a companion to the patients [13]. Another example is the NaoTherapist SAR [14], a rehabilitation instructor for patients with physical impairments.

In general, the morphologies or representation of the SARs usually vary depending on the application domain where they are used [15]. For example, robots used in mental healthcare applications vary their morphologies according to the required roles and functions of the robots, including zoomorphic, mechanistic, cartoon-like or humanoid representation among others.

### 2.1.1. Socially Assistive Robots for Older Adults

Concerning elderly care, the functionalities of SARs are related to the support of independent living by supporting basic activities (eating, bathing, toileting and getting dressed) and mobility (including navigation), providing household maintenance, monitoring of those who need continuous attention and maintaining safety [16]. There is therefore this dual use and perception: (i) social robots can be perceived as utilitarian systems: they are able to perform tasks such as housekeeping; and (ii) social robots are recognized as hedonic systems: they offer sociable interaction opportunities to be able to build long-term relationships with their users [7].

Fong and al. [17] had extensively described "socially interactive robots". They distinguish these robots—for which social interaction plays a key role [18]—from other robots that involve "conventional" human–robot interaction, such as those used in teleoperation scenarios. They can be useful for a variety of purposes: as research platforms, as toys,

as educational tools, or as therapeutic aids. The common underlying assumption is that humans prefer to interact with machines the same way they interact with other people.

This study focused on robots that exhibit the following "human social" characteristics: express and/or perceive emotions; communicate with high-level dialogue; learn/recognize models of other agents; establish/maintain social relationships; use natural cues (gaze, gestures, etc.); exhibit distinctive personality and character; may learn/develop social competencies. Their observation is that, although socially interactive robots have already been used with success, much work remains in order to increase their effectiveness. For example, in order for socially interactive robots to be accepted as "natural" interaction partners, they need more sophisticated social skills, such as the ability to recognize social context and convention. For them, the challenge is to build robots that have an intrinsic notion of sociality, that develop social skills and bond with people, and that can show empathy and true understanding. At present, such robots remain a distant goal, the achievement of which will require contributions from other research areas such as artificial life, developmental psychology and sociology.

Nearly twenty years after this survey [17], the challenge remains. In her article promoting multidisciplinarity, this cognitive psychologist's observation [19] was that after years of research in artificial intelligence and robotics, endowing technical systems with functionalities similar to human cognition and behavior still represents a scientific challenge. A holistic, user-centered, autonomous, and fully functioning robotic system that is capable of learning and of providing meaningful assistance and social companionship remains difficult to spot.

### 2.1.2. Human-Centered Human–Robot Interaction Design

The approach associated with the terms "human-friendly robots" [20], "human-friendly robot design" or "human-centred robotics" [21] appears to be being strongly technology-driven, and not that human-centered. The International Workshop on Human-Friendly Robotics defines it as "safe and dependable machines, operating in the close vicinity to humans or directly interacting with them in a wide range of domains". Developing human-friendly robots [20] rests on two key components: (i) smart interfaces and (ii) safe mechanisms—to ensure that people are never harmed. The latter aspect—high integrity safety systems that guarantee human safety by preventing dangerous impacts with people and the environment—is definitely essential (even more so with older adults, especially if the robot is intended to help mobility).

The main study of robots for older adults, which appears to be the most interesting as regards our own research interests (functions, use, robot's appearance, interaction modalities), is the Nursebot project [22,23]. Nursebot is a mobile robotic assistant, developed to assist older adults with mild cognitive and physical impairments, as well as support nurses in their daily activities. This nursing-assistant robot, named Pearl and its predecessor Flo, could provide many services, aiming at improving residents' quality of life. The two main researched services were: the task of reminding people of events, guiding them through their environment (accompanying residents is usually achieved by nurses, and this task is time-consuming).

The research team were aware of the necessity for the robot to adapt to the individual, an aspect of interaction that, then, had been poorly explored in AI and robotics. Aiming to address these challenges, three relevant software modules were developed to ensure successful human–robot interaction: an automated reminder system; a people-tracking and detection system; and finally, a high-level robot controller which performs planning under uncertainty by incorporating knowledge from low-level modules, and selecting appropriate courses of action. Systematic experiments—whether at the retirement home or in-lab—focusing on the robot's effectiveness from these software perspectives revealed successful tasks' performance. The tests and experiments adopting a quantitative approach, even when evaluating HRI (completion, error rate), not much is said about older adults except that the experiments allowed the authors "to gauge people's initial reactions to

the robot" [23] or that "Post-experimental debriefings illustrated a uniform high level of excitement on the side of the elderly" [22,23].

### 2.1.3. Contribution Linked to Literature Review

In their review of the studies of social robots [16], the robots examined in these studies range from "service type robots" providing functional assistance, to "companion type robots" providing affective assistance, with sometimes crosschecks between the two categories [16]. One of the failings pointed by this review is the methodological shortcomings. However, the limits are that the majority of studies (i) were performed with Aibo and Paro robots, and no other type of assistive robot, considering that form and material does matter a lot to the acceptance and effects of assistive social robots; (ii) were conducted in Japan: robot perception is culturally dependent, therefore results should not be simply generalized to other cultures; (iii) were done in nursing homes, preventing the generalizability to older persons living independently at home; and (iv) lacked methodological rigor: the research designs were considered as insufficiently robust, usually not described in enough detail to repeat, and confounding causal variables which could not be excluded [16].

On the contrary, analyzing the social acceptance of social robots in the home context, [7] point to the necessity of long-term studies (to prevent the novelty effect—which ends around two months of use—and because people's perceptions towards robots, their behaviors and their experiences are likely to change over time), in real environments, with a sufficient number of users. This is what the authors did in this research around co-designing a social assistive robot to support comprehensive geriatric assessment in a hospital. Taking all these previous work into consideration, and advocating for a human-centered approach, our use case of a robotic assistant for CGA procedure, which is a very pragmatic task-oriented type of human–robot interaction, is presented next.

### 2.2. CLARC Robotic Framework: Robot + Application

CLARC is an autonomous robotic solution to support CGA that is able to efficiently interact with the patient to gather data. During the tests, the CLARC robot collects, saves and displays the responses. It offers four interaction modes—vocal, tactile, gestural, physical buttons—thus adapting to the needs and preferences of each user. The CLARC robot is associated with a clinicians' application called CGAMed, that allows the physician to monitor the tests online or to access their results once the test is finished. Two geriatric tests were implemented: the Barthel test and the get-up-and-go test.

From a conceptual perspective, CLARC can be divided into two differentiated subsystems [24]. First, the *cognitive subsystem* is mainly focused on the robot's autonomy and its interaction with the patient. Our research, to date, has focused on designing efficient and acceptable human–robot interaction, by involving users in the design process [5], or understanding if and to what degree patient–robot interaction may influence the accuracy of robotized geriatric assessments (submitted), or the automation of one of the motion tests—the get-up-and-go test [25].

The second subsystem, the *CGAMed subsystem* (Figure 1a), focuses on the interface with the clinicians and the Clinical Data Management System (CDMS). In a typical CGA session, the clinician uses the CGAMed interface to setup the tests to be performed. Then, the robot conducts the tests autonomously and will store the results in the CGAMed database. Finally, the clinician uses the CGAMed interface again to review the session outcomes in order to create a new care plan.

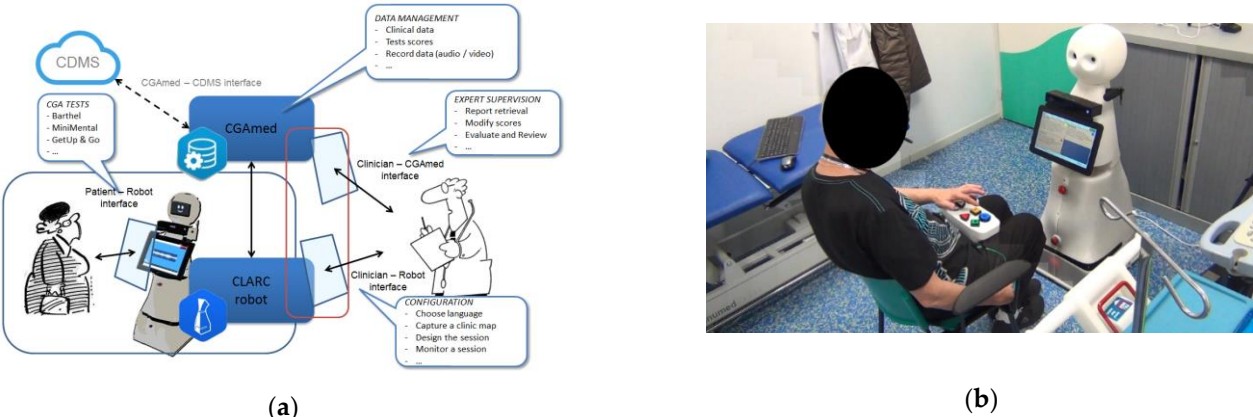

(**a**)                                                                                        (**b**)

**Figure 1.** (**a**) Global overview of the CLARC framework; and (**b**) real patient interacting with robot.

## 3. Methodological Approach of Fieldwork

Aiming at empowering the older adult, the approach adopted throughout this project is a Living Lab approach, combining iterative, human-centered and participatory design: users—both older patients and professionals—were involved throughout the iterative design and evaluation phase (of which two steps are described in this paper). "Living lab" is a concept created to support user-centric information and communication technology development processes. A living lab is defined both as a physical environment and an approach. This approach is useful in an innovation process towards the development of use scenarios in lab conditions, and real-life experiments in real-life homes, workplaces or healthcare settings. The approach consists of involving users in the design of future technological tools and services (Figure 1b). Its purpose is to design a simple implementation of the technology and ensure its usability and acceptability, considering not only the human–computer interaction but also the environment and context of use in which this interaction takes place [26].

### 3.1. Living Lab Approach

In our understanding and practice of the living lab approach, there are three important aspects:

### 3.1.1. Human-Centered Design

Human-centered interaction design would be the human–computer interaction centered on the exploration of new forms of living in and through technologies that give primacy to human actors, their values and their activities. The older adults and their needs are at the center of all research and design considerations. Human-centered design takes human (older adults) capabilities as a starting point, with a focus on how to support, develop and extend people's capabilities through the latest technological developments, in the domain of assistive technologies [27]. Moreover, following this approach, the traditional user-centered, four-stage design/research model—study–design–build–evaluate—was extended. A new fifth stage, which integrates at any point in the iterative design process, provides a framework to guide design and research. It entails conceptual analysis or "understanding" at the very beginning (Figure 2).

The extended approach to HCI (Human Computer Interaction) research and design is intended to enable human values to be folded into the mix at all the various stages. Harper et al. [28] give this example to explain what is meant by "values": one might be interested in developing new digital tabletop applications. This phase of work would involve clarifying what kinds of human values might be made possible through such interactions. Is it about supporting social connectivity and togetherness? Is it about play and creativity? Is it about archiving photographs and other materials to preserve and honor family history? Is it about allowing people to reminisce or reflect on their personal past? Or

perhaps is it about supporting collaborative tasks in domestic situations? Ultimately, this new stage of the cycle therefore results in making choices. It will also involve specifying what kinds of people are the focus of this particular project, and in what kinds of domains of activity, environments or cultures. In other words, it will involve choosing the kinds of value systems we are interested in. In the fieldwork described in this paper, it was about understanding the value systems of CGA in a healthcare setting, starting with basic research questions: who is involved in CGA, how is it organized, what data are being collected, why are the data important in the follow-up and when are they mobilized?

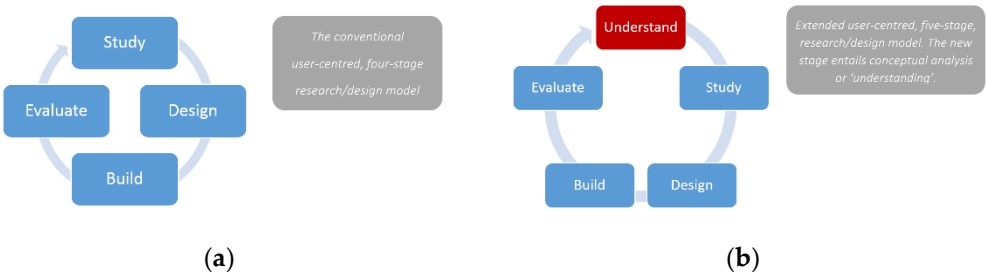

(**a**)                                                                                            (**b**)

**Figure 2.** (**a**) Conventional user-centered model; and (**b**) extended user-centered model following [28].

Together with the "study" stage, this understanding stage allows a relevant user study, using ethnography. The difference between this kind of analysis compared with the canonical HCI approach is that while a typical HCI project might only look at an individual's interaction or set of tasks or practices around a particular technology, the extended study stage can be much broader. This begins by considering the details of particular tasks or practices, but then asks how those mechanics of interaction help people achieve long-lasting value through and beyond the interaction. Research might look at current shopping practices, for example, and focus on how they enforce social connections to other people, or help people acquire new objects to bolster their identity, or how the shopping experience provides distraction and disengagement from the world of work (*ibid*) (p 31). It is precisely this kind of holistic understanding of organizational practices and culture in healthcare settings around data work for CGA that was aimed at the analysis of practices done in this fieldwork. As the authors explained, this extended approach also means talking to stakeholders—including users as well as those involved in developing or designing the technology in question—to ascertain what kinds of enduring value they believe their users will get from their technology; and what kinds of users and what domains are of interest. This is where this focus, on putting the human users and their values at the center of the design process, meets the principles of participatory design.

### 3.1.2. Participatory Design

Participatory design is a cooperative design process, with a focus on enabling different stakeholders with different perspectives and competencies to cooperate. It comprises active user involvement and participation in the design of IT artefacts and systems they will use, including in professional settings, where it is largely and increasingly used. Designers invite future users to participate in all phases of the design process [29]. Participatory design is generally united by an ethos of empowerment and "meaningful" involvement for stakeholders in the design of the systems they will use.

Participatory design has traditionally been useful in the design of technology applications or the co-realization of a more holistic socio–technical bricolage of new and existing technologies and practices. Moving away from the traditional computer and "user" notion, e.g., with ambient assisted living technologies [26] or social robots, there is indeed the need for participatory design. This co-design approach is useful in the ageing context to adapt solutions to the real-life situations of seniors. Indeed, it is often difficult for users, especially for seniors, to express their expectations and needs just by imagining them during interviews. Co-design, for example in dedicated workshops, provides a framework

and methodology for the emergence, expression and collection of these needs [29]. In addition to the pragmatic dimension (achieving an adequate design), the socio–political dimension is also important: the "voice" of future users, who are at the center of design concerns, is heard and taken into account and they have real decision-making power in the design and evaluation that is made of this technology. The notion of empowerment is therefore both an object of reflection, and also an objective to be achieved. It has guided the development of the AUSUS framework [paper submitted]—comprising the criteria Accessibility, Usability, Social acceptance, User experience, Societal impact—which has been used for the evaluation during the field trials (described in the field trials section, Section 6.1).

### 3.1.3. Practice Based: Ethnography

By "design", we do not only mean the design of human–robot interaction, but a global reflection about the work and organization in which this robot would be introduced. Central to this study is the notion that the design, use and evaluation of a mobile social robot for doing CGA cannot be done without serious attention being given to the organizational and work context [30].

Ethnography, performed as part of this fieldwork, has this ability to describe a social setting as it is perceived by those involved in the setting. In particular, it offers the opportunity to reveal needs or practices of users, which they themselves may not attend to, because they take them so much for granted that they do not think about them. In other words, these 'needs' cannot be articulated by the users themselves, because of either the bureaucratic or power relationships within which they are placed, or simply because they are too busy. This inability to articulate "needs" is, from the authors' experience, even more true of dependent older adults, especially those suffering from cognitive impairment. Instead of doing traditional in-depth interviews, which could prove dangerously unreliable, doing an ethnography allowed to gain an in-depth understanding of the "constitutive practices of how people do what they do, the 'interactional what' of their activities" [31].

### 4. Fieldwork: Analyzing Practices and Needs to Inform Design

Since its early days, CSCW (Computer-Supported Cooperative Work) has been concerned with healthcare, studying how healthcare work is collaboratively and practically achieved, and designing systems to support that work. The field has contributed to providing rich insights about healthcare practices, the use of paper- and electronic-based documentation, and the pinpointed implications for how to design collaborative systems. In their review of 25 years of CSCW research in healthcare, [32] argue that output from this research suggests that, apart from being complex and diverse, the challenges of implementing new technology in healthcare settings is locally situated. It is precisely this situated character of work that, following [33], we were interested in understanding.

### 4.1. Analyzing Practices

The main insights relate to the extent to which the creation, accumulation, management, and communication of data is central to patient work and to clinical work, but also to the management and governance of healthcare providers. Indeed, our research and design objective was not limited to the design of human–robot interaction, but was to achieve an in-depth understanding of the work and organization setting in which this robot would be introduced [4]. As explained earlier, central to this study is the notion that the design, use and evaluation of a mobile social robot (as for computer systems traditionally in CSCW), cannot be done without serious attention being given to the organizational and work context [30]. Therefore, the ethnography allowed to gain an in-depth understanding of the constitutive practices of data work based on the authors' conviction that "appropriate design" rests on an interest for the real needs and concerns of the central people involved [27]. Participant observation of CGAs and situated interviews of clinicians at their workplace were combined to understand work practices. The concept of practice [34]

refers to a regularly recurring activity, as actually unfolding, but also the "skills and competencies required to perform the activity in an accountable manner under conditions of contingency". Part of the professional skills and competencies, which emerge as important, are related to patient data management (detailed infra) and served as important insights to inform the robot's design and check the adequacy of the robot's functionalities with the organizational context and existing practices.

### 4.2. Skills and Competencies of Data Work

Part of the skills and competencies, which emerge out of our observations as important, are related to:

- Patient data requesting—*in* and *through* interaction with the patient;
- Patient data collection—as information are written down, either on paper or on the computer;
- Patient data retrieval—from the system, which reveals essential in the follow-up involved in CGA;
- Patient data interpretation—based on professional knowledge, but also knowledge of the organizational practices of the hospital, or usual personal practices of doctor X or doctor Y;
- Patient data sharing, either in a written form or to support in situ face-to-face collaboration between different clinicians involved in CGA.

Therefore, we see in the case of CGA in this day care center, how the creation, accumulation, management, and communication of data is central to patient work, clinical work, but also to the management and governance of healthcare providers. Explaining her practices to the ethnographer, the geriatric nurse naturally mentions that she inputs all the information "in the computer" as part of an organizational requirement. The main insight is the omnipresence of paper-based, together with electronic-based documentation. Is the "Paperless Office" [35] still a "myth" in the contemporary e-health context?

### 4.3. Insight: Paper- and Electronic-Based Documentation

Often, information appears on both paper- and electronic-based documentation. However, the nurse explains that the CGA tests themselves, i.e., the paper form with the coded information written on it, only appear in paper patient record. Depending on the test, e.g., a Barthel test which is questionnaire-based, it is only the result, most often a score and a simple comment, which is inputted in the system. For an MMSE (Mini Mental Score Evaluation), the psychologist writes a letter-like report, which is inputted in the system in pdf form. This pdf form is also printed and filed in the patient record.

Complementary Uses and Collaboration Practices

The nurse explains that the paper- and electronic- based information are complementary in their uses (Figure 3). She only has access to patients' paper files when the patient is admitted to the hospital for the day. Before the patient arrives, she makes the request for the patient file to the archives. When the patient leaves, she sends the record back to the archives. Therefore, in the patient's absence, the clinicians do not have access to the paper record. Whereas the electronic record can be accessed anytime, anywhere in the hospital, by any authorized clinician. The nurse explains that this allows her a quick and efficient patient information consultation. However, the limitation is that a paper patient record is internal to the hospital. If the patient has been admitted in another hospital, there is not only, no information, but no knowledge at all about the possible existence of another record in another hospital.

The practice of follow-up is also made easier by the affordance of paper. For comparing two CGAs made at different moments, the clinician can hold two sheets of paper and make an efficient comparison. Unless one has two screens or one very large screen, it is more difficult to compare two documents on screen. Moreover, since CGA is an interdisciplinary effort requiring the coordination of several clinical professionals, defining the appropriate

care plan is often a collaborative process. During the meeting, the referring doctor presents the case, and their colleagues can consult the record by reading the relevant document, which can pass from hand to hand. The observations are full of occurrences of collaboration between two or three clinicians standing in the consultation room around one sheet of paper, which supported the impromptu collaboration.

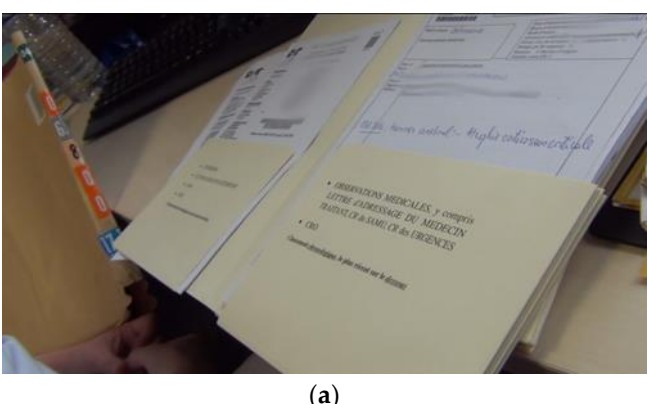
(**a**)

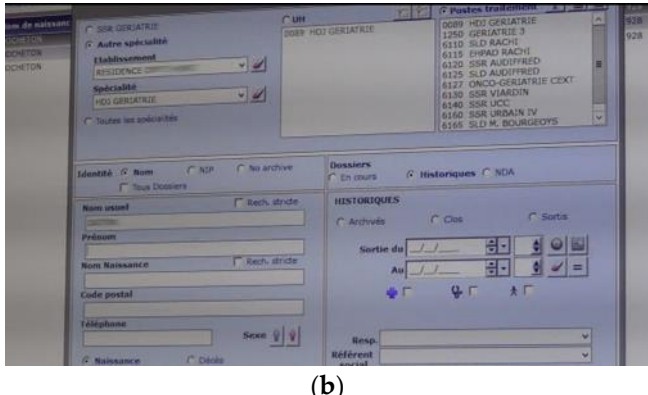
(**b**)

**Figure 3.** (**a**) Paper patient record on desk, next to keyboard; and (**b**) screenshot of electronic system.

### *4.4. Insights for Design*

This detailed analysis of data work has been useful to check the adequacy of the envisaged CGAmed—the clinician's application for data saving and data management— as regards existing practices of CGA, and the needs for data management in healthcare settings. Our research and design objective not being limited to the design of human–robot interaction (HRI), but to achieve a holistic adequate design of the robotic solution though an in-depth understanding of the work and organization in which this robot would be introduced [30]. The hypothesis underlying this fieldwork is that "adequate design" can only be achieved through considering the real needs and concerns of the central people involved. The participant observation of CGAs made by clinicians in the hospital context, as well as the situated interviews of clinicians at their workplace (hospital day care centers) were combined to understand work practices.

The main insight is the centrality of data work, and its complexity. The subsection above has described the advantages and drawbacks of paper-documentation. While paper documents afford in situ co-present collaboration, e.g., during collaborative diagnoses, where sheets can be passed from hand to hand, placed side-by-side for comparison, paper documentation can also seriously limit collaboration between healthcare providers. Therefore, as regards the current existing practices that have been observed in this organizational context and of the understanding of their complexity, the digitalization of data with CGAmed raises important and complex questions.

### 5. Methodological Approach to Field Trials

After several iterative loops of evaluation and improvements (Figure 4), the robot of level TRL8 (technology readiness level, meaning the system is complete and qualified) was tested in real-world conditions, in a rehabilitation hospital. This experiment combines the characteristics and objectives of both field trials [36,37], as practiced in the field of human–computer interaction, and those of pilot studies in health research [38,39]. Thus, it was possible to test this experimental system "in the wild", i.e., in uncontrolled real-life situations, in a complementary manner to the usability tests carried out in the previous stages [4,40], while evaluating the feasibility of the study in view of the next longitudinal study of the robot in a nursing home [41].

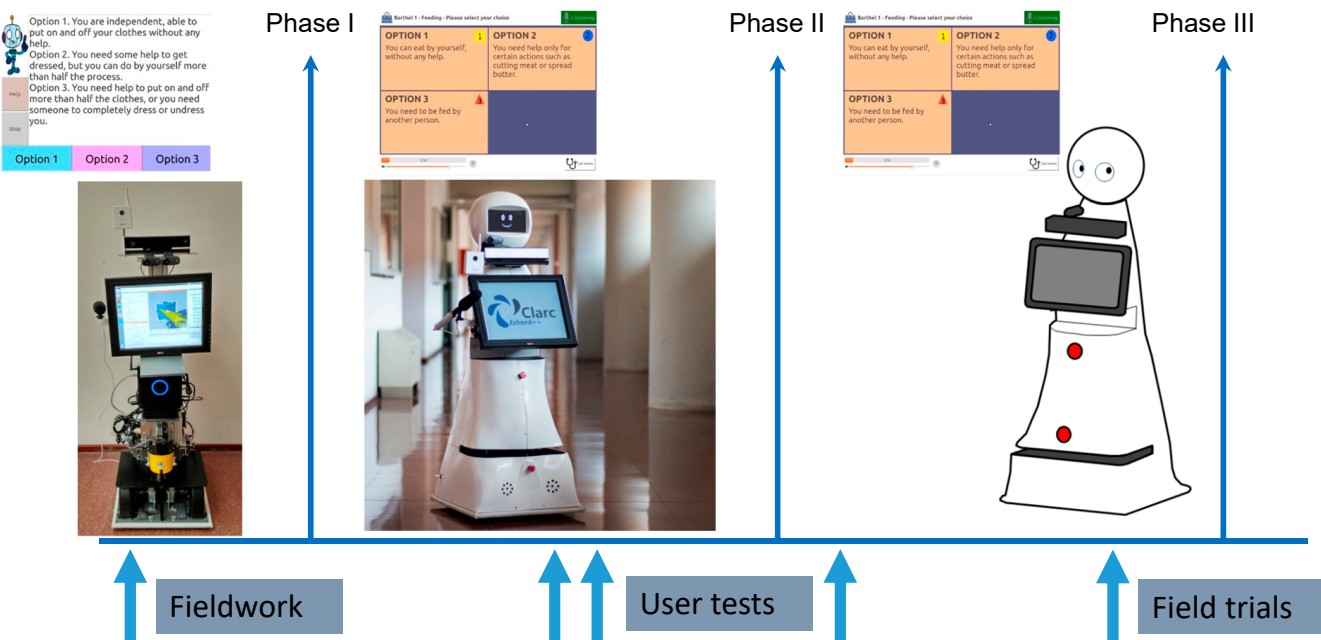

**Figure 4.** Iterative phases of robot evaluation and improvements.

These hypotheses, in terms of organizational objectives, were combined with the more practical evaluation aspects of human–robot interaction. Indeed, in order to be able to conduct these tests in an autonomous way, the CLARC robot must present itself to the older patient as an assistant that is accessible and useful. Older adults undergoing geriatric assessment, more than anyone else, are certainly not familiar with robotic technologies. In addition, age implies declines in motor, visual, hearing, and/or cognitive abilities. It is therefore crucial for the CLARC robot to be able to put them at ease and reassure them, and to offer them natural, intuitive, but also accessible modes of interaction [41].

### 5.1. Empowerment in and through Interaction

As in the methodological approach, the issue of empowerment, linked to the interaction of older users with the CLARC geriatric assistant robot, was taken into account in the design through two aspects. The first one was the interaction design. As described above, the CLARC robot offers flexibility in the interaction modes in order to adapt to the user's needs (according to their abilities/disabilities or use characteristics) and/or their interaction preferences.

Four different interaction modes were implemented in the CLARC robot—vocal, tactile, gestural, physical buttons—thus adapting to the needs and preferences of each user in an intuitive and accessible way [42]:

- The vocal mode was implemented through a text-to-speech (TTS) system, where the robot was able to talk to the patients, explaining what to do, making questions, etc. and the patients were able to talk to the robot and their speech was recognized by an Automatic Speech Recognition (ASR) system. This interaction mode implemented was one of the most natural ones in the robot. However, it was the least preferred by the patients [5].
- The tactile mode was implementing through the use of a touch-screen tablet situated in the torso of the robot. The different CGA tests were implemented in the tablet with questions to be answered by the patients and cognitive tasks to be performed, as the task of writing their name, memorizing terms or calculating among others.
- The gestural mode was implemented through the use of the touch-screen tablet, but also with the use of a video camera which detected the position of the patients in front. The video camera was able to detect, for instance, whether the patient was sitting, standing, walking, or even swaying too much and was in danger of falling.

- The physical button was implemented through a device external to the robot: a remote control that integrates a tablet and big physical buttons to help the patients complete the CGA tests from a more comfortable position than approaching the robot to press his tablet on his torso. They could do it from further away, sitting and without forcing their posture. Indeed, the remote control was developed during Phase 2 (see Figure 1b), based on the observation (results of successive user tests in France and Spain) that many users had significant difficulties with voice or touch control. In addition, the CLARC robot communicates verbally by programming the instructions of the synthetic voice so as to mimic a human attitude of benevolence towards its interlocutor: the least directive possible, allowing the necessary time of comprehension on the part of the user. In this way, the hypothesis examined is that the CLARC assistance robot would manage to put at ease users unfamiliar with robotic technologies, and to reassure them.

In addition to multimodality, the second aspect related to the empowerment of users in the interaction with the CLARC robot is the accessibility criterion. As previously discussed, age implies decreases in motor, visual, hearing, and/or cognitive abilities. The modes of interaction must therefore be natural, intuitive (aspect 1) but also accessible (aspect 2). The patients with hearing impairments, for instance, could not hear the robot's voice properly, but the multimodal interface allowed them to see the subtitles of everything the robots was speaking and moreover, the touchscreen in the robot's torso shows the task to do next or questions to answer next, so patients with hearing impairments could interact with the robots. Another example is that of patients with motor disabilities, e.g., patients with voice or hand tremors. The multimodality allowed them the use of the remote control with big and physical buttons to interact with. Moreover, the time to answer was extended according to the users' needs. Patients with visual impairments could use the voice to communicate with the robot. Moreover, they could use their own devices to connect to the robot (keyboard or mouse) and use Screen Readers to interact with the application because it was constructed as an accessible application following the recommendations of the WCAG 2.0 (Web Content Accessibility Guidelines) from the World Wide Web Consortium [43]. Finally, the cognitive disabilities and the impairments related to a lack of short-term memory were also taken into account, building the remote control in line with the touch-screen application (using icons, same shapes, colors, look and feel, etc.) and using simple sentences to communicate with the users during the whole interaction.

*5.2. Contribution: New Analytical Framework*

Following the multimodal interaction design, an analysis framework was specifically developed for this research project (Iglesias et al., forthcoming b.). In order to address the specific characteristics of geriatric patients, the accessibility criterion was found to be complementary to the other usual assessment factors. Accessibility, linked to the concepts of "universal design" or "design for all" [44] in HCI, has the main objective of producing systems that can be used by all, without discrimination, regardless of their physical or cognitive abilities [45].

The usability, social acceptance, user experience, societal impact (USUS) framework [46], a commonly accepted existing framework, allows the evaluation of humanoid robots used in collaborative tasks. The acronym corresponds to the following factors— usability, social acceptance, user experience, societal impact. Although representing a useful basis, the USUS framework proved to be insufficient for this study, due to the profile of users and the context of use (one-shot test). We therefore expanded the existing USUS framework to include the Accessibility criterion. Our contribution is the AUSUS analysis framework, a mixed methodological framework combining qualitative and quantitative approaches to evaluate human–robot interaction for older adults. In this pilot evaluation, AUSUS thus allowed us to assess the performance of the robot in performing the two tests—Barthel and get-up-and-go—while examining the effectiveness of the human–robot interaction (see details below, Section 6.2 Results of Field trials).

### 5.3. Mixed Methods Approach

This experiment combined the characteristics and objectives of both field trials [36,37], as practiced in the field of human–computer interaction, and those of pilot studies in health research [38,39]. Thus, it has been possible to test this experimental system in an uncontrolled real situation, in a functional re-education and rehabilitation hospital, in a complementary way to the laboratory usability tests carried out in the previous stages. Thus, the duration of the study—3 months (May to July 2019)—made it possible to prevent the novelty effect [6]. Analyzing social acceptance of social robots in the home context (*ibid*) point to the necessity of long-term studies, in real environments, with a sufficient number of users, to prevent the novelty effect. This novelty effect, they explain, ends around two months of use, because people's perceptions towards robots, their behaviors and their experiences are likely to change over time. Moreover, this pilot study allowed to evaluate the feasibility of the study for another longitudinal study, in continuity with this research, which started in November 2019 in Andalusia [41].

Aiming at the complementarity of approaches, the authors adopted a mixed methodological approach [47] in the research protocol. Defined as integrating the collection and analysis of quantitative and qualitative data in a single study or program [48], the complementarity it brings is advocated [49]. The objective of using this mixed methodological framework is two-fold: on the one hand, triangulation, which ensures the validity of quantitative analyses [50], and on the other hand, the complementarity of quantitative and qualitative methods, the latter allowing for a more refined and contextualized understanding of quantitative results [49]. The objective is to produce results that combine credibility and meaning [51].

Mixed methods research has been of interest to health researchers for many years. Various combinations of methods have been used in health research in the context of service evaluation, the exploration of health issues, and the development of research instruments [47]. They are coming into health research practice at the same time as more and more professionals and researchers are convinced of the usefulness of qualitative methods [51], including in France.

Thus, the AUSUS analysis framework, developed specifically for this research, is mixed quantitative (durations, measurement of criteria according to a Likert scale) and qualitative, combining an interest for what people *say* (interview) and for what people *do* (ethnographic observation, analysis of activity by video recordings), throughout the course of the test.

### 6. Field Trials: The Pilot Evaluation

The results presented briefly in this paper are based on the Barthel test with a cohort of 25 patients. As defined by our consortium partners in the ECHORD++ CLARC (the European Clearing House for Open Robotics Development) project (No. FP7-ICT-601116), the University Hospital of Seville, the inclusion criteria were: aged 65 years or older; MMSE test score $\geq$ 23. Exclusion criteria were: hearing impairment; visual impairment. The hypothesis was that cognitive, hearing and visual abilities are the necessary conditions for human–robot interaction. Indeed, in order to interact efficiently with the robot, the patient should have the cognitive abilities to adapt to the robot and understand the instructions, and should be able to read the written text on the screen and hear the text-to-speech from the robot's speakers.

The Barthel test is a 10-question questionnaire on functional abilities. For this test, the patient-user can answer using three different modalities: vocal interaction, tactile interaction with the screen or by pressing the buttons on the remote control. While the movement recognition is important for the other test implemented on the robot, the get-up-and-go test, the gesture modality was not used for this specific test described in this paper.

### 6.1. Test Procedure and Protocol

The test is organized as follows. First, the study is presented by the investigator: the patient signs the informed consent after being fully informed. Then, a short pre-test questionnaire is used to determine the level of use of technology by the patient-testers, their impressions of robots, and their experience with the geriatric evaluation, in particular the Barthel test. The test itself is videorecorded (three complementary angles, see Figure 5)), following video ethnography principles [52], for a detailed *a posteriori* activity analysis, and observed in real time by the investigator, who remains in the room with the patient and the robot (without intervening).

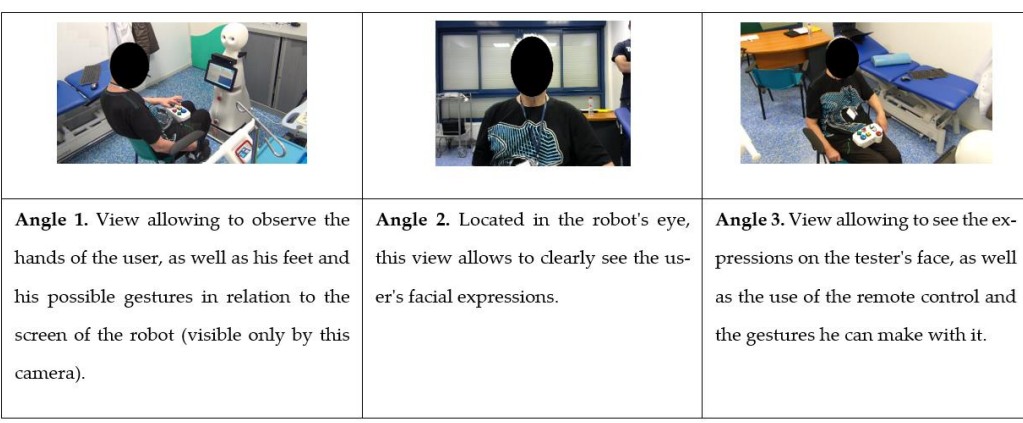

**Angle 1.** View allowing to observe the hands of the user, as well as his feet and his possible gestures in relation to the screen of the robot (visible only by this camera).

**Angle 2.** Located in the robot's eye, this view allows to clearly see the user's facial expressions.

**Angle 3.** View allowing to see the expressions on the tester's face, as well as the use of the remote control and the gestures he can make with it.

**Figure 5.** Different camera angles to capture and analyze the activity.

After the test with the robot, the AUSUS questionnaire guides a semi-structured interview, during which the investigator can deepen certain aspects of interaction in a way that is relevant to the in situ observation he has just carried out.

### 6.2. Results

The main insights presented in this section relate to: performance, interaction factors, test duration, interaction mode.

The system proved to be robust, with an acceptable technical performance rate: 24 patients out of 25 completed the test, which means that there was one failure out of the 25 tests made. Moreover, as explained above, the AUSUS analytical framework was used for the evaluation of performance indicators. After the robot interaction session, during a semi-structured interview based on a questionnaire, patients were asked to evaluate the interaction aspects on a 5-point Likert scale ranging from 1 (strongly disagree, lowest score) to 5 (strongly agree, highest score).

The different aspects of interaction that were evaluated received encouraging scores (Figure 6): an average above 4 out of a total of 5 (learning; flexibility; perceived usefulness; social acceptability; robot physical aspect; robot perception; safety), and above 3 out of 5 for the more complex aspects of human–robot interaction (concentration; emotions).

With the exception of three patients, whose test durations were below or above average (9 min 10, 15 min 35 and 14 min 21), the average duration of the tests was 10 min (12 tests) or 11 min (10 tests) (Figure 7).

Concerning the interaction mode, the remote control emerges as the preferred mode of interaction (Figure 8). This preference confirms the intermediate hypothesis that emerged from the user tests in phase 2 of the project, namely the need for an interaction mode, other than voice and touch, that would be easier and more accessible for older users.

Finally, all patients reported being satisfied by having participated in this study. In line with the living lab approach adopted throughout this project, especially the participatory dimension, this feedback is insightful for our future research.

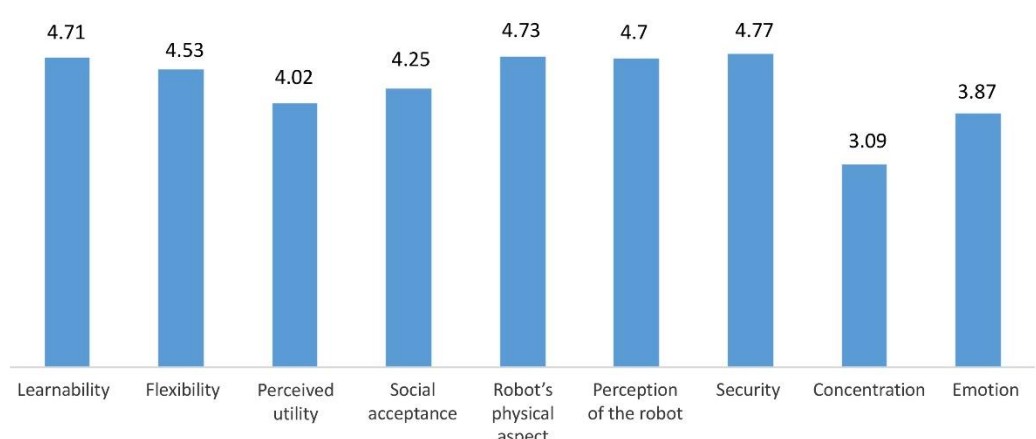

**Figure 6.** Evaluation scores of interaction aspects (out of /5).

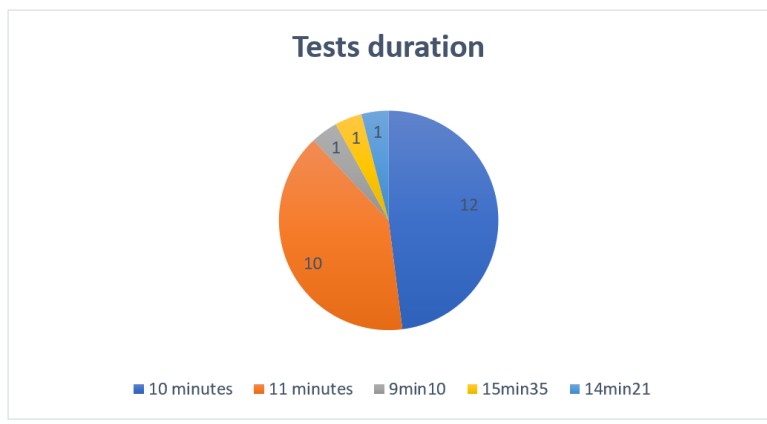

**Figure 7.** Average test duration: 10–11 min.

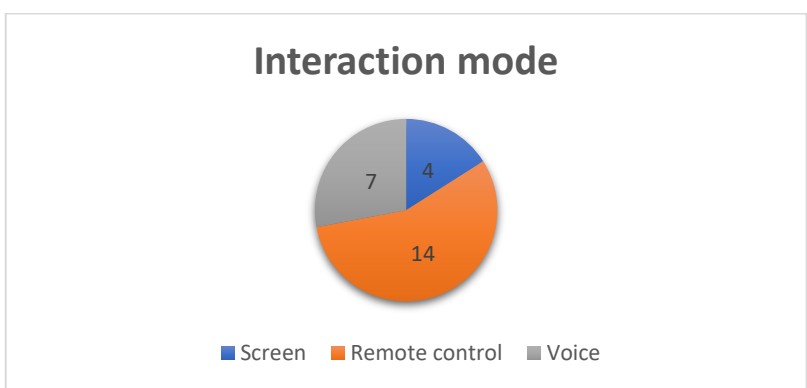

**Figure 8.** Different user preferred modes of interaction.

## 7. Discussion

The knowledge gained—first from this field work to understand current data work practices, and second from this real-life pilot evaluation—has been used to inform the design of the robot. It is part of an iterative improvement concerning the interface, voice interaction, intelligibility of instructions, and motion detection.

More specifically, from a pragmatic perspective, the field trials allowed to check the adequacy between existing workplace practices and the functionalities of this new technology that would be introduced in the healthcare settings. The ethnographic observations allowed

to understand the organizational context of data work—collection, saving, sharing in CGA patients' follow-up. The utility of such a robotic solution and the functions of CGAmed application was confirmed. From a reflexive perspective, CGA proves to be an interesting case study for a global reflection about the place and role of data work in supporting collaboration between people, and among different healthcare organizations. The relevance of information collected in CGA within the global objective of ageing well and autonomous living at home, questions the types of data that need to be collected, e.g., in the design of Ambient Assisted Living technologies, and how robotics and Artificial Intelligence can contribute to it. Our study is also an interesting case study to advance social robotics applications in healthcare settings for older adults, where the applications so far (cf. literature review section) have mainly focused on nursing homes and domestic environments.

Concerning the field trials, thanks to the duration of the experiment (3 months), the novelty effect (2 months) [6] was avoided. Even if the actual use during the test was one-shot, the regular presence of the robot at the rehabilitation center allowed a certain degree of familiarization. Moreover, a form of social acceptability seemed to appear in the ethnographic data through the positive attitude of most of the patients and staff of this hospital center, regarding the presence of the CLARC robot, and concerning the hypothesis of a gain in efficiency for CGA procedure. From a pragmatic and design-oriented perspective, the knowledge gained allowed to inform the continuous improvements of the robot in terms of movement detection, human–robot interface, and voice recognition. Indeed, despite the promising evaluation results succinctly presented in this paper, human–robot interaction remains a complex question. Its use and interaction design—and certainly acceptability—reveals to be much more complex than other ambient assisted living technologies in which we have been involved in to date, be it connected balance scales to detect physical frailty [53], a falls detection device [54] or virtual personal assistants to prevent older adults' social isolation [55]. For example, voice interaction, despite its theoretical "naturalness", is the least preferred by the patients. The system was not robust enough, certainly due to the increased difficulties of recognition of older adults' speech [56] and we are trying to improve it with other text-to-speech and automatic speech recognition systems.

Knowledge gained from these field trials was also used to evaluate and improve the AUSUS framework, which was used in this pilot evaluation, with a view towards its generalization. In particular, this experimentation confirms the value of a mixed methods approach, combining a quantitative approach, allowing an objectivization of the results, and a qualitative approach, allowing the results to be fully meaningful. The interest of this approach is confirmed in health research, in HCI, and all the more so, for an object as complex and new as human–robot Interaction. From a methodological and reflexive perspective, the knowledge generated by this pilot experiment also serves to feed the reflection on the future deployment of such a fully functional technological tool in a longitudinal clinical study (ROSI and ITERA projects which constitute the follow-up work to ECHORD++ CLARC, cf. [41]. ROSI: Robotic assistants for nursing homes; ITERA: Integration of Assistive Robotics Technologies in residences for the elderly). The difficulties observed linked to the complexity of this "one-off use" situation suggests the hypothesis that further familiarization (with actual use) is key to efficient human–robot interaction, especially for older adults. The next step of our research, in line with some of the authors' previous work with another robot [18], is to examine the use and appropriation of a social robot in nursing homes.

## 8. Conclusions

Based on an extensive literature review of social robotics applications for health and ageing, this paper has discussed the practical and methodological questions raised around how to design and test assistive social robots for clinical routine. These field trials demonstrate that an automated CGA procedure is actually feasible, with promising results in terms of performance and user satisfaction. However, the real efficiency of an automating

CGA procedure, including in terms of data collection, test duration and user experience, needs to be further examined in a longitudinal study in actual use (v/s pilot studies).

**Author Contributions:** Methodology, A.I., K.L.H.T. and D.V.; software, R.M.; formal analysis, K.L.H.T.; investigation, D.V. and K.L.H.T.; resources, R.M.; writing—original draft preparation, K.L.H.T.; writing—review and editing, D.V. and A.I.; supervision, Q.D.R.; funding acquisition, D.V. and A.I. All authors have read and agreed to the published version of the manuscript.

**Funding:** This research was partially funded by the EU ECHORD++ project (FP7-ICT-601116), the TIN2015-65686-C5-1-R Spanish Ministerio de Economia y Competitividad project and FEDER funds, the CSO2017-86747-R Spanish project, the ITERA project (UMA-FEDERJA-074), the ROSI project (AT17_5509_UMA), and by Berger Levrault as part of the company's robotics research delegation to the ActivAgeing Living Lab.

**Institutional Review Board Statement:** The study was conducted according to the guidelines of the Declaration of Helsinki, and following the recommendations of the UK Medical Research Council (MRC ethics series - Using information about people in health research, Version 2.0, June 2018) by the Board of COS CRRF Pasteur, March 4, 2019. It complies with the requirements of the General Data Protection Regulation (GDPR) concerning data protection law applied to research.

**Informed Consent Statement:** Informed consent was obtained from all subjects involved in the study.

**Acknowledgments:** The authors warmly thank Guillaume Beghin, intern at ActivAgeing Living Lab; Jean-Luc Novella; the team of the former geriatric day care center of the Centre Hospitalier de Troyes; colleagues from the EU project ECHORD++ CLARC; colleagues from CRRF COS Pasteur at Troyes—especially Guillaume Dessinger, Mattieu Detroy, Philippe Voisin; and most of all, the patients of CRRF COS Pasteur, for their participation in this research.

**Conflicts of Interest:** The authors declare no conflict of interest.

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
