# Peer review of "Fieldwork and Field Trials in Hospitals: Co-Designing A Robotic Solution to Support Data Collection in Geriatric Assessment"

_applsci, doi:10.3390/app11073046_

Round 1

Reviewer 1 Report

1. The manuscript describes the feasibility of utilizing information technology to facilitate comprehensive geriatric assessment in the clinical setting.

2. As human subjects are involved, information about an IRB review is required.

3. The manuscript is written primarily as a description of the methodology and justification of the approach utilized.  The manuscript will be more useful to clinicians if the focus is placed on the technical performance and human interface experience. The authors should state clearly in the introduction that the manuscript represents a report on the feasibility of an automated comprehensive  geriatric assessment procedure, describe the development of the concept, design of the user interface and protocol, followed by reporting the clinical testing and user experience. Barriers, limitations, and next iterations should be detailed.

4. The tense should be in the present, with avoidance of future connotation: "are coming into",  "would be", etc.

5. Some grammatical and wording suggestions are added as comments to the attached manuscript.

Author Response

Dear reviewer,   

first of all, we would like to thank you for your constructive comments, which allowed us to significantly improve the paper.    

> The manuscript will be more useful to clinicians if the focus is placed on the technical performance and human interface experience.   We focused it more in the introduction and made sure this is clear in the analysis part.  

> The authors should state clearly in the introduction that the manuscript represents a report on the feasibility of an automated comprehensive geriatric assessment procedure.   This has been done in the Introduction and has been further discussed in the Discussion section.  

> Barriers, limitations, and next iterations should be detailed.   It has been further developed in the Discussion section.  

> The tense should be in the present, with avoidance of future connotation: "are coming into", "would be", etc.   The present tense has been used when possible, e.g in the use scenario which has been changed from future to present tense. However, "in which the robot would be introduced" is conditional tense, and we will stick to it, because beyond this pilot experimentation, the robot has not been deployed for actual use in real situations.  

Thank you very much again for your review, which has been very useful, and also for taking time to read this rebuttal letter.  

Kind regards,   

Karine Lan Hing Ting & co-authors

Reviewer 2 Report

Dear editor,

I have read and reviewed the article entitled „Fieldwork and field trials in hospitals: Co-designing a robotic solution to support data collection in geriatric assessment”. I think that the article presents an important topic for the field, namely, how to design and test assistive social robots for clinical routines, and it might be interesting for the readers of the Journal. However, I also think that's the article needs major improvements before considering it to be published in the journal „Applied Sciences”.

I have several main concerns. First, I think that the article has a poor introduction and state of the art analysis. I think that the authors should describe more in detail previous work that has been done in terms of design and evaluation of social robots that are being used as robotic assistants in medical contexts, perhaps for similar purposes. They should also delineate what are the advancements that are being proposed by their work in the context of such previous developments.

I also think that they describe in many details the general philosophy of participatory design (section 3 in the manuscript) and spend less space for describing their work for the current study. I suggest that the introduction of the article should be followed by a brief and focused presentation of the development of their solution.

Third, I find that many of the hypotheses that are being put forward in the manuscript are very hard to test and are barely being addressed by the data and study presented in this manuscript. For example, there is no data about the following (main) hypothesis: „a robotic solution would allow a gain in efficiency in geriatric follow-up”.

Fourth, I think there are several concepts that are being named in the manuscript, while their meaning is a little unclear. For example, they discuss remote control as being the preferred method of interaction, without explaining what this type of interaction is about and to what other types of interaction have been compared to. The presentation of the items that were used to assess the perception of the interaction with the robot needs more details. For example, what did they assess with the items concerning concentration and emotions?

There are also some smaller details that should be attended to. For example, Figure 6 on page 11 is in French, and just below this figure, there is a repetition of the word „below” which makes the phrase unclear.

Finally, the discussion section should be rewritten in order to reflect what can be learned from this experimentation, comparing to previous studies and examples of assistive technologies offered by the literature.

I think that the article should be reconsidered for publication after the revisions I have suggested above.

Author Response

Dear reviewer, 

first of all, we would like to thank you for your constructive comments, which allowed us to significantly improve the paper.

We answer each issue you raised below.

> First, I think that the article has a poor introduction and state of the art analysis. I think that the authors should describe more in detail previous work that has been done in terms of design and evaluation of social robots that are being used as robotic assistants in medical contexts, perhaps for similar purposes. They should also delineate what are the advancements that are being proposed by their work in the context of such previous developments.

Our focus was on describing the empirical work done, and discussing the methodological issues raised, and admit we have been wrong in neglecting the literature review. Following the Reviewer's 2 comment, we significantly improved the state-of-the-art, with an extensive literature review, positioning our contribution as advancement or in line with previous work (Literature review and Discussion sections).

> 2. (...) I suggest that the introduction of the article should be followed by a brief and focused presentation of the development of their solution.

With enriching the literature review, the text has been reorganized. The hypothesis has been integrated in the Introduction, resonating with previous work, therefore strengthening the introduction in terms of the focus and contribution of the paper

> Third, I find that many of the hypotheses that are being put forward in the manuscript are very hard to test and are barely being addressed by the data and study presented in this manuscript. For example, there is no data about the following (main) hypothesis: "a robotic solution would allow a gain in efficiency in geriatric follow-up”.

Indeed, this paper is more about a methodological discussion and reflexive discussion of the research, and not so much about confirming the hypothesis. Indeed, presenting it as a "hypothesis", making it a title (like in the previous version) can create these false expectations. 
We hope we succeeded in mitigating the scope of this hypothesis in the discussion of the limitations in the Discussion section, and that with the restructuring of the hypothesis presentation as a broad research question in the Introduction (rather than a dedicated section), the reader does not expect a "confirmation of the hypothesis" as a logical second part in the rest of the paper. 

> Fourth, I think there are several concepts that are being named inthe manuscript, while their meaning is a little unclear. For example,they discuss remote control as being the preferred method ofinteraction, without explaining what this type of interaction is about and to what other types of interaction have been compared to. 

We have detailed further the different interaction modes, and explained in detail the remote control.

> The presentation of the items that were used to assess the perception of the interaction with the robot needs more details. For example,what did they assess with the items concerning concentration and emotions?

Due to the focus of the paper (discussing the methodology rather than presenting the results in detail), and the significant enriching of the Literature review following the Reviewer 2 suggestion, the paper would definitely be too long if we make this suggested change. The factors being assessed are described in detail in another paper, cited in this one (Iglesias et al.? (to be published)).
So, if Reviewer 2 agrees, this section will not be modified.

> Finally, the discussion section should be rewritten in order to reflect what can be learned from this experimentation, comparing to previous studies and examples of assistive technologies offered by the literature.

The Discussion section has been improved in this way, discussing this case study in light of previous social robots research, and other AAL projects.

Thank you very much again for your review, which has been very useful, and also for taking time to read this rebuttal letter.

Kind regards, 

Karine Lan Hing Ting & co-authors

Round 2

Reviewer 1 Report

  1. The authors have made a credible response to reviewer comments.
  2. The revised manuscript has greatly enhanced clarity and organization.
  3. Please add a conclusion to the paper and revise the last sentence of the abstract (line 27) as a conclusion statement - perhaps related to feasibility.
  4. Do not use "elderlies" rather "older adults" as the accepted age-friendly term. Examples:
    1. line 49 - older adults
    2. line 243 - consider "empowering the older adult"
  5. Other line corrections:
    1. line 161 – promoting
    2. line 188 – delete either "imperative" or "necessity" - do not use both terms
    3. line 270, no upper case in the abbreviation, use either e.g. or i.e.
    4. line 322 - reflection

Author Response

Dear Reviewer 1, 

thank you very much for your careful reading and final improvements suggestions, which have been all taken into account. 
I detail them below.

> Please add a conclusion to the paper and revise the last sentence of the abstract (line 27) as a conclusion statement - perhaps related to feasibility.
The requested conclusion related to feasibility has been added.

Do not use "elderlies" rather "older adults" as the accepted age-friendly term. Examples:
line 49 - older adults
line 243 - consider "empowering the older adult"
We have changed into "older adults everywhere in the text.

> Other line corrections:
line 161 – promoting --> line 152
line 188 – delete either "imperative" or "necessity" - do not use both terms --> line 177
line 270, no upper case in the abbreviation, use either e.g. or i.e. --> line 258
line 322 - reflection --> line 310

All these changes have been done

Thank you again for this very useful review.

Best regards, 

Karine & co-authors

Reviewer 2 Report

I think the papers is much more readable and clear now. The introduction covers relevant studies and offers a good context for their work. Also, the fact that the idea of several hypothesis to be confirmed by the data has been removed has moved the focus to the qualitative dimension of the study which offers a good presentation of the the entire process of developing and testing the utility of the robotic agent.

Author Response

Dear Reviewer 2,

Thank you very much for your careful reading, and your comment, which confirms that the paper is now relevantly improved and ready for publication.

Best regards,

Karine & co-authors